# Irrigated Crop Types Mapping in Tashkent Province of Uzbekistan with Remote Sensing-Based Classification Methods

**DOI:** 10.3390/s22155683

**Published:** 2022-07-29

**Authors:** Elbek Erdanaev, Martin Kappas, Daniel Wyss

**Affiliations:** Cartography, GIS and Remote Sensing Department, Institute of Geography, University of Göttingen, Goldschmidt Street 5, 37077 Göttingen, Germany; daniel.wyss@uni-goettingen.de

**Keywords:** crop types mapping, Sentinel, Landsat, SVM, RF, NDVI, EVI, NDWI, OSS data, irrigated land, Uzbekistan

## Abstract

Appropriate crop type mapping to monitor and control land management is very important in developing countries. It can be very useful where digital cadaster maps are not available or usage of Remote Sensing (RS) data is not utilized in the process of monitoring and inventory. The main goal of the present research is to compare and assess the importance of optical RS data in crop type classification using medium and high spatial resolution RS imagery in 2018. With this goal, Landsat 8 (L8) and Sentinel-2 (S2) data were acquired over the Tashkent Province between the crop growth period of May and October. In addition, this period is the only possible time for having cloud-free satellite images. The following four indices “Normalized Difference Vegetation Index” (NDVI), “Enhanced Vegetation Index” (EVI), and “Normalized Difference Water Index” (NDWI1 and NDWI2) were calculated using blue, red, near-infrared, shortwave infrared 1, and shortwave infrared 2 bands. Support-Vector-Machine (SVM) and Random Forest (RF) classification methods were used to generate the main crop type maps. As a result, the Overall Accuracy (OA) of all indices was above 84% and the highest OA of 92% was achieved together with EVI-NDVI and the RF method of L8 sensor data. The highest Kappa Accuracy (KA) was found with the RF method of L8 data when EVI (KA of 88%) and EVI-NDVI (KA of 87%) indices were used. A comparison of the classified crop type area with Official State Statistics (OSS) data about sown crops area demonstrated that the smallest absolute weighted average (WA) value difference (0.2 thousand ha) was obtained using EVI-NDVI with RF method and NDVI with SVM method of L8 sensor data. For S2-sensor data, the smallest absolute value difference result (0.1 thousand ha) was obtained using EVI with RF method and 0.4 thousand ha using NDVI with SVM method. Therefore, it can be concluded that the results demonstrate new opportunities in the joint use of Landsat and Sentinel data in the future to capture high temporal resolution during the vegetation growth period for crop type mapping. We believe that the joint use of S2 and L8 data enables the separation of crop types and increases the classification accuracy.

## 1. Introduction

Land use or land cover maps are the primary tools to manage information on the Earth’s surface and the interaction between different land cover types. In this context, the need to distinguish between land cover (i.e., the physical properties of land surface) and land use (i.e., human activities making use of land) needs to be mentioned [1], with this paper focusing on land use, although it is not possible to identify land use without identifying land cover. Up to the 1990s, most information on land use in Uzbekistan was derived from national mapping and surveying programs with standards of spatial as well as thematic resolution widely varying between countries and global regions [2]. While the capacity for land use mapping at a global scale had been steadily developing, starting with the first Landsat satellite in 1972, it was the opportunity to make global RS imagery more widely available via what was initially the internet and later became the world wide web that data, as well as processing capabilities and more affordable image processing software, became available to a wider range of users globally [3]. In addition to their use in land management, land use maps are also of relevance in the context of environmental objectives such as the land use, land use change, and forestry sphere within the area of climate change politics and research [4] or biodiversity research [5,6].

Before the advent of RS technology, traditional cadaster data formed the basis of national land use statistics. Initially, national cadastral mapping had been introduced for taxation reasons as a basis for land ownership records as well as land use distribution [7,8]. However, in some countries of the world including the study area, adequate information is not yet available. This is due to a historical lack of cadaster systems as well as an outdated land management system. These issues related to the distribution of power between transnational (e.g., European Union), national and sub-national entities, with the latter a particularly vexing factor in countries with federal rather than centralized administrative and political systems [9].

RS data have been used widely in the field of crop phenology. Aside from crop identification, this includes the identification of crop growth stages. Usually, very coarse resolution (i.e., 100–250 m) products such as MODIS (Moderate Resolution Imaging Spectroradiometer) and VIIRS (Visible Infrared Imaging Radiometer Suite) have been widely used and proven helpful in landscape and regional level yield prediction. However, for analysis at field scale, medium (i.e., 30 m) to high (i.e., 10 m or better) resolution products, especially the Landsat series of sensors, are considered more beneficial. Unfortunately, in many regions of the world, the lack of clear (cloud-free) imagery poses a considerable challenge to widespread operational use [3]. The availability of RS imagery and relevant processing capabilities have led to a wide range of uses of RS data in agriculture. Common applications include crop growth and yield assessment, irrigation research, and information on potential crop losses due to pests and diseases [10]. Similarly, agricultural land use monitoring and crop yield forecasting are listed as the main application areas [11]. The recent interest in precision farming as an approach to improve economic efficiency, reduce negative environmental impacts through minimizing the use of herbicides and pesticides, and avoid over-fertilization has led to the development of multi-data-source approaches incorporating artificial intelligence (AI) technology. Frequently noted examples include the use of higher spatial resolution data gathered from conventional aerial sensor technology as well as sensors for the visible spectrum and passive hyperspectral sensors but also active light detection and ranging (LIDAR) or radar sensors. Other approaches include training AI with a combination of high or medium solution RS data (e.g., Landsat or Sentinel) to allow for information of higher accuracy to be derived from low-resolution RS sensors (e.g., MODIS), which provides the benefit of more frequent return periods [11,12].

The most cultivated agricultural crops are maize (corn) with global annual consumption of 1107 Mt, wheat with global consumption of 740 Mt, and rice with an annual consumption of 510 Mt [13]. Consequently, considerable work has been conducted in the context of RS methods to improve information on these crops. The following research works provide a short review related to our research work. For maize, research on land use suitability for cultivation in Indonesia was carried out by [14], using NDVI as well as SAVI based on L8 data to assess cultivation potential, although in this study the areas identified had not yet been cultivated with maize. Zhang et al. demonstrate that with appropriate methods it is possible to not just identify maize as a crop but to improve phenological analysis to allow for the differentiation between common maize (i.e., for human or animal consumption or industrial purposes) and seed maize [15]. The phenological analysis was based on different vegetation indices (VI), including NDVI, EVI, triangle vegetation index (TVI), ratio vegetation index (RVI), NDWI, difference vegetation index (DVI), and an RF classification. Furthermore, they also assessed classification accuracy using the KA and producer accuracy (PA). Satellite RS-based maize acreage estimation and prediction of maize yields based on a combination of L8 NDVI and land surface temperature (LST), data as predictors were analyzed in work published by [16] using ground-truthing plots for supervised classification. These results are also of particular interest due to more frequent extreme weather situations, due to climate change, expected to lead to higher variations in yield.

Climate change and phenological reactions of agriculture crops were also the main focus of research carried out by [17], focusing on the spring phenology response of winter wheat to pre-season weather data based on long-time climate records as well as NOAA-AVHRR NDVI time-series data from 1981 to 2015. Results demonstrated the approach to be more suited to assess more long-term climate change developments rather than short-term seasonal reactions and also highlighted the potential of using long-term time series data available from historical RS records. However, Wang et al. arrived at different results using the spring frost damage index to identify spring frost damage using a combination of historic weather data and MODIS RS data [18].

Given its importance as a staple crop in many of the most populated regions of the world, considerable work has also been carried out using RS data in the context of rice farming. For the identification of rice, different classification methods including supervised and unsupervised classification as well as phenological indicators have been used successfully with newer work focusing on AI approaches. For yield prediction, sophisticated methods need to be used, given the required differentiation between irrigated, rain-fed, or upland paddy fields. Given these complexities, additional data such as digital elevation models (DEM) need to be used. For more precise crop yield prediction, complex crop models using additional input, e.g., canopy height from LIDAR data (manned aircraft or UAV) are required [19].

To summarize, the use of RS for identifying not just the most important crops but also for analyzing phenological development details within farmed areas with these crops has been widely used with recent approaches combining RS and machine learning (ML) technology.

Classical unsupervised classification is based on a statistical analysis of natural groupings of data, typically using cluster-based approaches to analyze the degree of similarity of data correlation between different bands within pixels. Classic supervised classification is based on the same principle but uses training areas to provide ex ante information on areas of different land-cover types. Training areas are classified either using ground-truth data or through visual interpretation by a human operator. The main problem with unsupervised classification is that spectral data will not always correspond to spectral classes and that the final grouping of clusters needs to be decided by the human operator. Supervised classification, on the other hand, provides more accurate classifications and also allows for more control over the classification process. Especially for large areas and diverse conditions (e.g., differing seasonality of phenology at different sea levels), extensive and thus expensive training is required [20].

In this context, the main goal of this research is to test the capability of using S2 and L8 sensor data for mapping precise and accurate main irrigated crop types using ML algorithms SVM and RF. To achieve this goal, the following specific objectives were developed: (i) to map and compare the performance of ML algorithms such as SVM and RF for main irrigated croplands by crop types with medium and high-resolution L8 and S2 data; (ii) to test different index combinations such as NDVI, EVI, NDWI1, and NDWI2 as input data to derive crop type classification; (iii) to compare the area of all derived agricultural land use maps with the OSS data from the State Committee for Statistics of Uzbekistan. The other parts of this research work are structured as follows: Section 2 describes the study area, presents the data description, and theoretical background of methodologies of ML classifiers for crop types mapping, Section 3 describes the results and provides a discussion of the results. Lastly, Section 5 draws some conclusions and the future direction.

## 2. Materials and Methods

### 2.1. Study Area

This study focuses on the Tashkent province within the central Asian country Uzbekistan. Agricultural land makes up for about 62% of the total land area, the majority of this being pasture, while only about 10% of the national land area of some 425 km^2^ is available as arable land [21]. Tashkent Province was formed in 1938 as part of the Uzbek Soviet Socialist Republic and is located in the northeast of Uzbekistan between 40.18 N and 42.29 N and 68.64 E and 71.27 E of the Greenwich meridian or between the western part of the Tien Shan mountains and Syrdarya river. The Province borders Kazakhstan in the north and north-west, with Kyrgyzstan in the north-east, Namangan Province in the east, Tajikistan in the south, and with Syrdarya Province in the south-western part [22]. Since arable land is located only in the lower areas of the province, the analysis only focused on these regions (see Figure 1).

“The climate is a typically continental climate with humid, relatively mild wet winters and long, hot, and dry summers. The mean January temperature is −1 °C to −2 °C and the mean July temperature is 26.8 °C. The average annual precipitation is 300 mm in the plains region, 300–400 mm in the piedmont region, and 500–600 mm in the mountains. Precipitation mostly occurs in the early spring and permanent snow cover is located in the higher mountains. The main river Syrdarya and its tributaries Chirchik and Akhangaron Rivers basins are fed by snow and glaciers and they are used for irrigation and hydroelectric power [22]”. In Uzbekistan, wheat is cultivated on about 40% of irrigated lands, cotton is around 36% and the remaining 24% is other crops (fruits, vegetables, livestock, and various cereals). In the Tashkent province, cotton and wheat occupied over 61% percent of the total cultivated area in 2018 [23].

### 2.2. Data

S2 and L8 tiles covering the relevant study region were downloaded from the USGS Earth Explorer [24] site for multiple dates of vegetation growth period from May to October 2018. In addition, only cloud-free images are available during this time. An overview of the satellite imagery dates for different months is provided in Table 1. In total, 4 tiles of S2 (T42TVL, T42TWK, T42TWL, T42TWM) and 3 tiles (153/031, 154/031, 153/032) of L8 were downloaded and processed separately before merging. All tile id numbers of S2 start with T (Toulouse) and the second two numbers 42 is the Universal Transverse Mercator (UTM) zone, next T is the latitudinal chunk, and the last two letters denote the position of the tiles. L8 tiles path and row numbers show the location of Tashkent province.

The specification of spectral bands for the two sensor systems used in the analysis can be observed in Table 2.

One of the significant problems in crop type mapping in this research is the lack of quality training data. Because of ideal ground reference data limitation, historical Google Earth images in 2018 (February, April, June, July, August, and October) available in Google Earth Pro desktop application (Google LLC, version 7.3.2.5776, Göttingen, Germany) was used as an alternative for validation pixel samples collection. It is based on prior knowledge of crop phenology and cropping calendar [27]. The spatial distribution of training and validation samples is displayed in Figure 2, and the numerical information is given in Table 3.

The OSS data [23] about the sown area by crop types across provinces of the country in 2018 issued by the State Committee of Statistics of Uzbekistan is used for comparative analysis with derived remote sensing-based crop types area.

### 2.3. Methodology

#### 2.3.1. Data Preprocessing

After downloading S2 imagery, the SWIR 1 and SWIR 2 bands were resampled to 10 m resolution. Both L8 and S2 data were then atmospherically corrected from Top of Atmosphere Reflectance (TOA) to Surface Reflectance (SR). This was conducted using the Dark Object Subtraction (DOS1) tool of the Semi-Automatic Classification Plugin (SCP) of the QGIS GIS package [28]. Then, all tiles of S2 and L8 merged for every month separately, and subset to study area. Reflectance values were then used to calculate NDVI [29], EVI [30], NDWI1 [31] from SWIR 1, and NDWI2 from SWIR 2. In the end, we have obtained monthly temporal profiles of NDVI, EVI, NDWI1, and NDWI2 as input data for ML classifiers.

Before classification built-up areas were manually digitized as polygons and the shapefile was used to mask these areas.

Training data was used to train SVM and RF classifiers. Both classifiers were then used to classify the main irrigated crop type maps. This was performed for 5 combination variants using different indices such as (1) NDVI, (2) EVI, (3) EVI-NDVI used together, (4) NDWI1, and (5) NDWI2 data.

The resulting land use maps were then assessed for accuracy using OA, PA, UA, and KA using validation data for reference.

#### 2.3.2. Indexes

Four spectral vegetation indices, NDVI [29], EVI [30], NDWI1 [31] from SWIR 1, and NDWI2 from SWIR 2 were calculated using the surface reflectance values. These indices were formulated by using the following equations:(1)$$NDVI=\frac{\rho_{NIR}-\rho_{Red}}{\rho_{NIR}+\rho_{Red}}$$
(2)$$EVI=2.5\times \frac{\rho_{NIR}-\rho_{Red}}{\rho_{NIR}+6\times \rho_{Red}-7\times \rho_{Blue}+1}$$
(3)$$NDWI1=\frac{\rho_{NIR}-\rho_{SWIR1}}{\rho_{NIR}+\rho_{SWIR1}}$$
(4)$$NDWI2=\frac{\rho_{NIR}-\rho_{SWIR2}}{\rho_{NIR}+\rho_{SWIR2}}$$
where ρ_Blue_, ρ_Red_, ρ_NIR_, ρ_SWIR1_, and ρ_SWIR2_ are the surface reflectance values of Band 2 (blue, 0.45–0.51 μm), Band 4 (red, 0.64–0.67 μm), Band 5 (near-infrared, 0.85–0.88 μm), Band 6 (SWIR1, 1.57–1.65 μm), and Band 7 (SWIR2, 2.11–2.29 μm) in the Landsat-8 LOI and Sentinel-2 images, respectively (Table 2).

The approach is displayed in Figure 3.

#### 2.3.3. ML Algorithms

SVM is a statistical learning method, which was first published by [32]. The SVM training algorithm is designed to identify a hyperplane separating a dataset into predefined discrete classes based on training examples. The decision boundary minimizing misclassifications is considered the optimal separation hyperplane. This is identified through an iterative learning process separating first training patterns and then simulation data with the same configurations [33].

In RS, individual pixels are represented as pattern vectors consisting of numerical measurements for each frequency band. In addition, other discriminative measurements based on spatial pixel relationships (e.g., texture) may also be elements of the feature vector. The hyperplane of maximum margin is defined by the subset of points lying on the margin of the classes. In Figure 4, the concept is illustrated as a linear SVM based on the simple example of a two-class classification problem. In RS practice, more complex SVMs are applied using multi-class classifiers as kernel functions. A major advantage of SVMs is that they also work well with small training datasets while achieving higher classification accuracy than conventional approaches [35]. Another advantage is the fact that SVM allows generalizing accuracy acquired from finite training patterns to unseen data.

The main challenge for the use of SVM in RS is constituted by the choice of kernel functions. In this context, the radial-bias function and polynomial functions have been demonstrated to produce different results [33]. In this research, we used a linear kernel function in which the algorithm creates a hyperplane to separate the classes.

The RF method was first introduced by [35]. It is based on a combination of tree predictors in which each tree depends on the values of an independently sampled random vector, where all trees in the forest have the same distribution. For the set-up of an RF model, the base of the method, constituted by the two parameters, the number of trees *n* and the number of features in each split *m_try_,* are required. According to [35], a random forests consists of tree-structured classifiers {h(x, Θk), k = 1,…}. In this {Θk} independent, there are identically distributed random vectors. Accurate classification is determined by each tree casting a unit vote. In RF classifiers, the number of features used at each node and the number of trees grown are user-defined parameters. Thus, at each node only selected features are assessed. In the classification of a dataset, each case is assessed in each tree. Accurate classification is determined by the majority vote from all trees [36].

The concept is illustrated in Figure 5. A are input samples. B and C are decision trees within an RF D, assigning the sample to one of two branches based on the rule at each decision point. In both B and C, the sample is assigned to the red class. Consequently, the combined output result E of the RF is also the red class. The RF has strong predictive performance. In addition, results inform each feature’s level in contributing to class prediction [37].

Comparing SVM and RF, ref. [37] concludes that they achieve comparable accuracy. The fact that RF only requires two parameters to be set, whereas SVM requires several user-defined parameters constitutes an advantage of RF over SVM [38]. RF classifier tool in ArcMap creates models and generates predictions based on Leo Breiman’s RF algorithm [35]. Another advantage of RF is the ability to handle data with missing values and unbalanced data, as well as categorical data, which SVM lacks. In addition, RF allows for the detection of outliers through proximity analysis. The main difference though is that RF can also be used for unsupervised classification. These advantages of RF not-withstanding, ref. [39], consider SVM with the polynomial kernel as well as radial-basis function to be superior to RF, with RF performing inferior to SVM, if only single satellite coverage is used.

#### 2.3.4. Accuracy of RS Classification

To assess the performance of each classifier with different satellite sensors and indices combination, the confusion matrices were calculated in ArcMap. Confusion matrices can be used to describe the classification algorithm’s performance.

The increased use of digital RS data and various semi-automated or fully-automated classification methods has led to an increased interest in classification accuracy. OA is the simplest statistic, describing the number of all correctly classified pixels by the total number of pixels used for accuracy assessment within the error matrix. In addition, producer’s accuracy (PA) describes the probability that a reference pixel has been correctly classified (and thus not omitted); the user’s accuracy (UA) describes the probability of pixels within a specific class, which have been correctly classified and divided by the total number of pixels assigned to that class. Another common measure of accuracy is KA, which is calculated by the kappa index of agreement KHAT equation [40]:(5)$$\hat{K}=\frac{N\;\sum_{i=1}^{r}x_{ii}-\sum_{i=1}^{r}\left(x_{i+}x_{+i}\right)}{N^{2}-\sum_{i=1}^{r}\left(x_{i+}x_{+i}\right)}$$
where:

*r*: *number of rows in the matrix;*

*x_ii_*: *number of observations in row i and column i;*

*xi_+_*, *x_+i_*: *marginal totals of row i and column i;*

*N*: *total number of observations.*

KA value can be divided into three categories; a value greater than 0.80 represents strong agreement; a value between 0.40–0.80 represents moderate agreement; and a value below 0.40 represents poor agreement [40].

## 3. Results and Discussion

### 3.1. Results from Land Use Classification

Results from the land use classification for different combinations of sensor, classifier, and index are presented in Figure 6 (S2-data and SVM classifier), Figure 7 (S2-data and RF classifier), Figure 8 (L8-data and SVM classifier), and Figure 9 (L8-data and RF classifier). For comparison in each figure the OSS data are also presented.

With winter wheat being the most abundant crop in the study region, classification results for this crop might have a higher influence on the overall result. Figure 6 shows that wheat results based on S2-SVM indices are very close to OSS and each other. Figure 7, however, shows that except for EVI-NDVI this is also true for S2-RF. From Figure 8, it can be observed that there are higher differences between the different indices as well as between individual indices and OSS. However, what is also evident from Figure 8 and Figure 9 is that the SVM and RF classifiers tend to classify rice paddies area with a probability of difference between 190% to almost 500% compared to the area given by OSS. This can be the result of wetlands being classified as rice fields and needs to be investigated further in detail. Besides, the rice is also planted after harvesting winter wheat as a second crop which is not included and recorded by OSS. This is because the study area is located where the upstream water resources are formed and it has more access to the water resources than other areas of the country.

As can be observed from Figure 9, there are differences between OSS data and classification results for each of the sensor-classifier-index combinations. These differences appear to be particularly high for land use classes for which OSS shows comparatively low values. To analyze this in more detail, differences between individual classification results and OSS were calculated and presented in Table 4.

For each classification method, the arithmetic means (AM), as well as a WA using OSS area as weights, were calculated. Since using the AM value for the differences, the results in a disproportionate influence of differences for land use classes with a low share of overall land use (i.e., particularly rice); therefore, it was decided to focus on the WA results, as shown in Table 4.

Based on the results, the smallest absolute value difference WA 0.1 thousand ha was demonstrated for the S2-sensor data using the RF classifier and EVI index, whereas for the SVM classifier with S2 data, the smallest absolute value difference WA result, 0.4 thousand ha, is obtained using the NDVI.

For L8-sensor data, the smallest absolute value difference result of 0.2 thousand ha was obtained by using the EVI-NDVI with RF method or NDVI with SVM method.

To compare classification results for different sensor-classifier-index combinations, subsets of land use maps in the middle of the study area located in Figure 10 for the SVM classifier and Figure 11 for the RF classifier. The location of subsets is shown in Figure 12 and Figure 13. The subset area was chosen due to its location along the main river Chirchik and the availability of all main cultivated crops inside. Visual analysis of these comparisons indicates that results appear similar for both sensors’ data with differences depending on the classifier and index method. The results from the NDWI2 demonstrate a very high amount of areas classified as water compared to other index methods.

Since OSS data are based on planning data compiled by the Ministry of Agriculture and Resources of Uzbekistan, Tashkent, Uzbekistan, rather than on an assessment of the actual cultivation situation, a high level of similarity for classification results with these data is not necessarily an indicator of the quality of the classification result. Therefore, results for classification accuracy are presented and analyzed in the next section.

### 3.2. Classification Accuracy Results

Classification accuracy (CA) was assessed using ground-truthing samples derived from historical Google Earth data. Results for OA, UA, PA, and KA are presented in Table 5 for the SVM classifier and in Table 6 for the RF classifier.

As can be observed from Table 5, UA and PA vary considerably between land use classes for the SVM classifier. The AM across all land use classes is in the range of 84% to 89% for all sensor-ML algorithm-index combinations. The highest mean UA value for the SVM classifier is 89% for S2-EVI-NDVI. The highest mean UA value for L8 is 83% for NDWI1 and NDWI2, followed by 82% for NDVI and EVI-NDVI.

Looking at Table 6 and using OA and KA as measures, these two classification accuracy indicators are higher for the S2 than the L8 sensor data using the SVM classifier. In comparison to this, when looking at Table 6 at OA for results from the RF classifier and S2 and L8, this relationship is less pronounced, with OA showing higher values for S2 than L8 for NDVI, and NDWI2, whereas OA for L8 has taken on higher values than S2 for EVI, EVI-NDVI, and NDWI1. KA shows the same relationships between sensor data for both SVM and RF.

Results presented in Table 5 and Table 6 indicate that RF with L8 data results in higher OA values than SVM with L8 data. However, for S2 data, results from SVM show higher OA values than with RF.

The highest OA of 88% resulted for S2 was achieved when EVI-NDVI was used, as well as the highest KA. A map displaying the classification result is shown in Figure 12. In addition, the highest OA of 90% for L8 data was achieved with EVI. A map displaying the classification result is shown in Figure 13.

For the RF classifier in Table 6, results for UA and PA also differ considerably between land use classes and sensors. AM values across all land use classes for each satellite datasets-ML algorithms calculation are in the range of 82% to 88%. The highest mean value of 89% resulted for EVI-L8, EVI-NDVI-PA-L8, and EVI-NDVI-UA-L8. The highest mean UA value for S2-data is 88% for NDWI2 followed by 87% for EVI-NDVI. The highest PA value is 83% for L8-NDWI2 at 83% and L8-EVI-NDVI. As can be observed from Table 5 and Table 6, OA is the highest, at 90%, for the combination L8-RF-EVI, followed by L8-RF-EVI-NDVI, at 89%, and S2-SVM-EVI-NDVI and L8-RF-NDWI2, both 88%. The values for KA, on the other hand, are highest for L8-RF-EVI at 88%, followed by L8-RF-EVI-NDVI at 87. Generally, the values for KA show a wider range from 75 to 88 than those for OA, which range from 84 to 90. The mapping result for the combination L8-RF-EVI is displayed in Figure 13.

## 4. Discussion

In this study, high-resolution S2 data, as well as medium-resolution L8 data were analyzed using different vegetation and water indices to derive main irrigated crop types mapping. Two widely used ML classifiers of SVM and RF methods were used to recommend appropriate classification methods to map high-resolution spatial crop types in the semi-arid area of Tashkent province, Uzbekistan, because SVM and RF classifiers perform better results for cropland classification when compared with other classification methods such as Maximum Likelihood Classification, Classification and Regression Trees, Naive Bayes, etc. [28,41].

### 4.1. Performance of ML Classifiers

Comparison of accuracy assessment analysis indicates that the highest OA 90%, as well as KA 88%, was achieved using L8 sensor data with RF classifier and EVI used as input data. For the SVM classifier, the highest OA 88% as well as KA 86% resulted from using S2 data and EVI-NDVI used as input data. Thus, regarding the accuracy assessment analysis, results from this paper do not provide a definitive answer on whether S2 or L8 is a better dataset for crop types classification, as the OA and KA result rather depends on the classifiers (SVM or RF) than the type of sensor (medium-resolution L8 or high-resolution S2) used. However, we can conclude that in terms of accuracy assessment, the RF classifier performs slighter better than SVM. This result also agrees with other studies [42,43,44,45].

Thus, the conclusion made by Nitze et al. [39] that SVM performs superior to RF if used with single satellite coverage is not supported by the results presented in this paper. In conclusion, the results demonstrate that both classifiers perform well, with comparable results in terms of classification accuracy, which are made in [35], to be supported.

### 4.2. Using Different Indices and Their Performance

Using the spectral indices alone improves the classification accuracy. When spectral indices are used together with reflectance values for crop type classification, it negatively impacts classification accuracy due to large sets of correlated variables. The reflectance including bands Blue, Red, NIR, SWIR1, and SWIR2 spectral indices are very useful for the identification of crop types and can achieve high classification accuracy [46]. Based on that recommendation, four indices of NDVI, EVI, NDWI1, and NDWI2 performance were studied using different satellite datasets and ML algorithms.

The highest values of OA and KA resulted when EVI and EVI-NDVI were used in crop type classifications, as shown in Figure 14 and Figure 15. Using the L8 dataset’s EVI, EVI-NDVI, and NDWI1 with RF classifier yielded higher OA values of 90%, 89%, and 87%, respectively. It also applies to KA values as well. The lowest OA and KA values resulted in an SVM classifier of L8 datasets for all indices.

For NDWI1 and NDWI2, a visual assessment of classified images in Figure 10 and Figure 11 indicates that these indices tended to overclassify water areas, which could explain the overall bigger areas classified to these classes because NDWI is very sensitive to changes in the water content of vegetation canopies and soil water content [31]. Due to irrigation of crops in the early crop growing period, the soil water content influences the classification accuracy, which resulted in many misclassified pixels as water. NDVI and EVI resulted in overall slightly better accuracy values than NDWI1 and NDWI2. Vegetation indices, which include NIR, have a great contribution to identifying crop types [42,47,48]. The other research studies compared the performance of EVI and NDVI for image classifications; it was found that their performance is equally good, with a slight overperformance of each other’s [49,50]. The most important factor is crop phenology knowledge and crop growth period consideration during training and validation samples [51]. At the very beginning of the crop growing season (March–April), the classification achieves relatively low accuracies, and it significantly increases when satellite images are obtained between May and June until the OA reaches its highest value in July [52,53]. However, when the combination of multiple sensor datasets is used for crop type classification, it improves the classification accuracy due to its high temporal resolution, which captures more datasets throughout the vegetation growth period [51,54,55].

### 4.3. Comparing Derived Main Crop Types Area with OSS Data

Cropland areas derived from this study were compared with OSS data at the provincial level. The other studies also compared calculated croplands area with OSS data at national and regional levels [27,56,57,58]. In Figure 16, a comparison of the total area classified for the land use categories of cotton, wheat, rice, and other crops is displayed by the sensor, classifier, and index and compared to the equivalent figures from OSS.

The comparison of the total area classified for the land use categories of cotton, wheat, rice, and other crops with OSS data in Figure 16 shows that with exception of the S2-RF sensor-classifier combination, classifications based on NDVI are closest to OSS and also comparatively close to each other regarding the total classified area. NDWI1 and NDWI2 also demonstrate similar total classified area results for all sensor-classifier combinations, but at roughly 250 thousand ha both show smaller total areas than OSS data. The total area of classifiers NDVI-S2-RF, EVI-S2-SVM, EVI-NDVI-L8-RF, and EVI-NDVI-S2-RF are recorded in OSS.

The mapped areas of the crop classes in this study area, overall, do well with OSS data at the provincial level, with on average 1% deviation in coverage resulting with L8 sensor datasets using RF-EVI and SVM-EVI-NDVI methods, as shown in Table 7. A similar result was also found in the work of Asam et al., 2022 [58]. The deviation between 1% to 13% resulted from the EVI and NDVI used by all ML classifiers and sensor datasets. The close values are also found in the research of Oliphant et al., 2019 [57]. The lowest deviation values of 8% to 14% (less than OSS data) were found using NDWI1 and NDWI2 by all ML classifiers and sensor datasets.

### 4.4. Theoretical and Practical Implications of the Research

Land information systems access for local or regional land administration highly influences the way a state operates and the policies they develop. Recognizing land uses by land administration is a major funding source such as tax, stamp duty on property transfers, etc. [43]. Derived irrigated crop types maps can be utilized by regional land administration offices to monitor the spatial extent of crops location and its monitoring as well as modeling and predicting crop yields and production by different models.

### 4.5. Limitations and Recommendations

Training and validation sampling points were taken based on the best knowledge of crop development phenology, cropping calendar, monthly NDVI profiles during the crop growth period, and historical google earth images. However, it is lacking the field observation data. In this research, we focused on creating a map of major crops such as cotton, wheat, rice, other crops, and fruits/trees, which are recorded by state statistics. The “Other crops” class consists of multiple minor crops, and we are motivated to continue our research to classify these classes by crop types in the future. Besides, we recommend creating detailed second crop maps, which is vital for public land management authority and local government to make decisions on a crop location that is suitable for soil quality and food security stability.

## 5. Conclusions

In this study, S2 and L8-based time series data in 2018 served as the input for mapping irrigated crop types using different vegetation indices such as NDVI, EVI, NDWI1, and NDWI2 by SVM and RF ML algorithms.

Regarding the comparability of medium (30 m) to high (10 m) resolution RS data, the results have demonstrated that both sensor products provide comparable outcomes concerning total area classified as well as accuracy assessment results. This is confirmed by the fact that the classification results demonstrating the highest OA results for SVM and RF, respectively, have been produced with different sensors. A closer analysis of OA, KA, UA, and PA, too, has demonstrated that RS imagery from both sensors is of comparable quality. Differences in accuracy results vary higher based on the vegetation indices used than on sensor data. KA values vary between 75% to 88% in all indices. The lowest KA values were achieved in all indices with the SVM classifier of L8 sensor data. The highest KA values of 88% and 87% were achieved with the RF classifier of L8 data when EVI and EVI-NDVI were used, respectively.

We can also have a similar conclusion regarding the difference between RS-based-derived crop types area and OSS area. The lowest absolute WA using OSS area as weight, between areas classified per respective OSS category is 0.1 thousand ha for S2-RF-EVI classification and 0.2 thousand ha for L8-SVM-NDVI classification. Thus, classified maps can be used by global cropland mapping projects or any other ecological models that require specific crop types mapping. The recent successful launch of Landsat-9 will successfully continue the Landsat data suite and enable new opportunities in the joint use of Landsat and Sentinel data to capture high temporal resolution during the vegetation growth period.

The comparison of classified map areas for the main crops of cotton, wheat, and other crops demonstrated very reliable results when NDVI and EVI-NDVI were used with S2 sensor data. Therefore, S2 high resolution and temporal data can be utilized in the future to create LU maps to calculate crop water and irrigation requirements by using a variety of climate-hydrology models. Besides, created crop types map of agricultural areas can be used to inform decision-makers of local administration offices to develop policies to assure food security, valuable ecological resources, and services where digital data do not exist or missing.

The performance of both ML classifiers with S2 and L8 sensor data resulted in not being satisfactory for the rice crop due to the mixture of this class with second crops after winter wheat, which was due to a similar sowing time and growth period. Another limitation of this research relates to other crops classes, where other crop classes include a variety of crops; some of these crops might have similar growth periods and spectral properties to each other. This can influence on the OA of the classifiers, and we recommend further studying the separating of other crop types in the future.

## Figures and Tables

**Figure 1 sensors-22-05683-f001:**
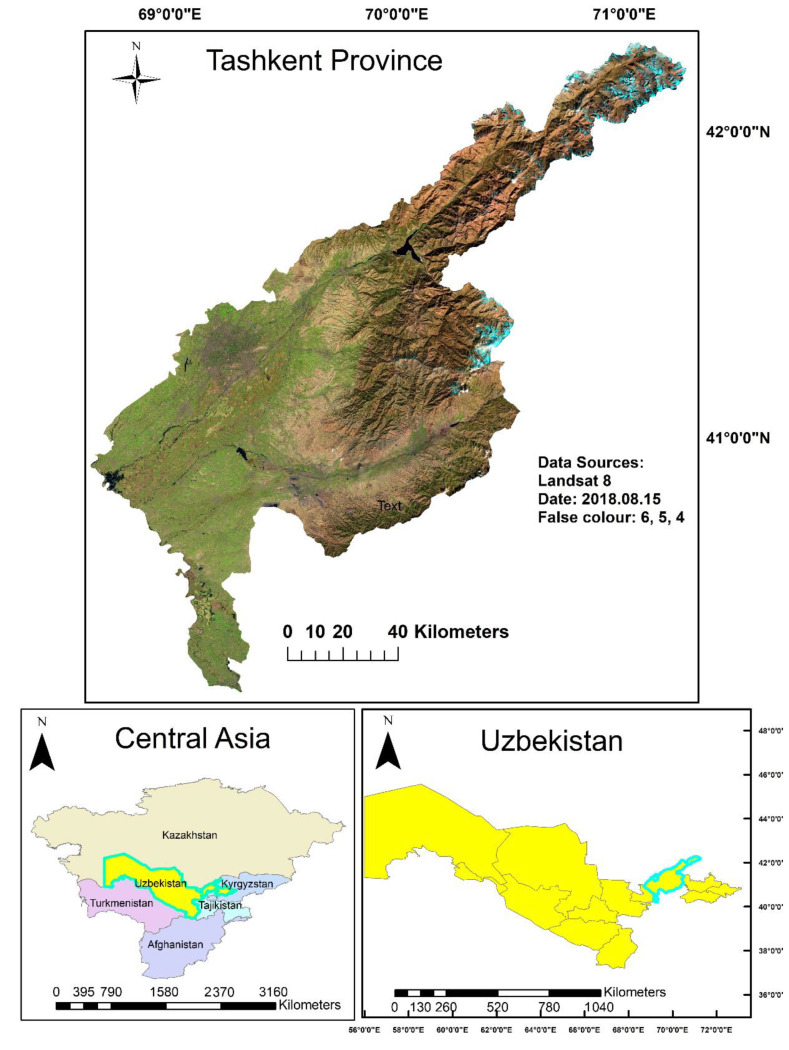
Location of the study region.

**Figure 2 sensors-22-05683-f002:**
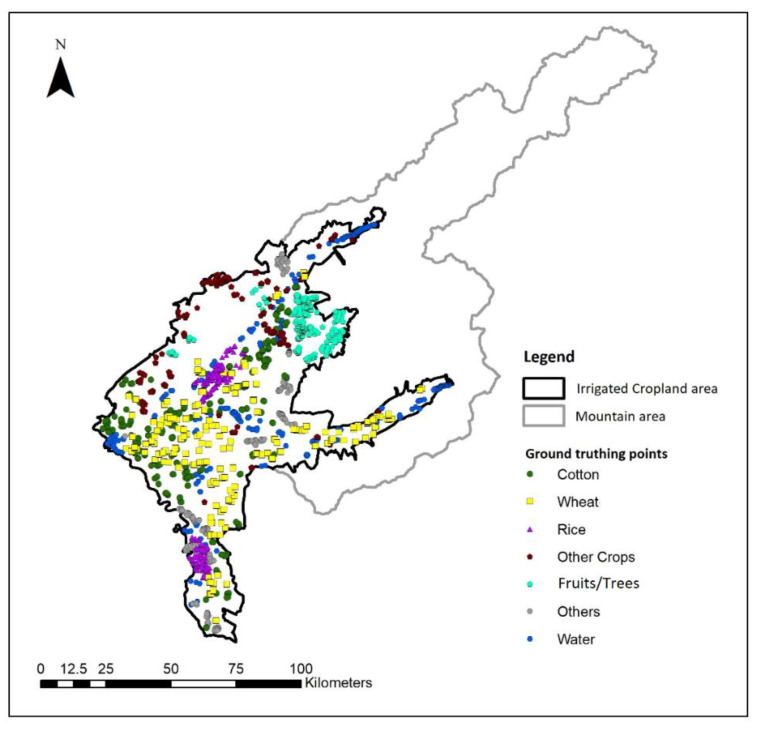
Location of training and ground-truthing samples.

**Figure 3 sensors-22-05683-f003:**
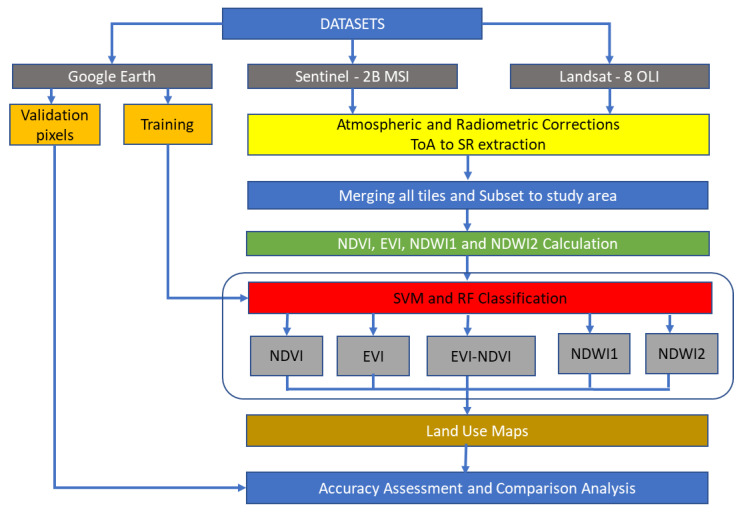
Methodology applied for this research.

**Figure 4 sensors-22-05683-f004:**
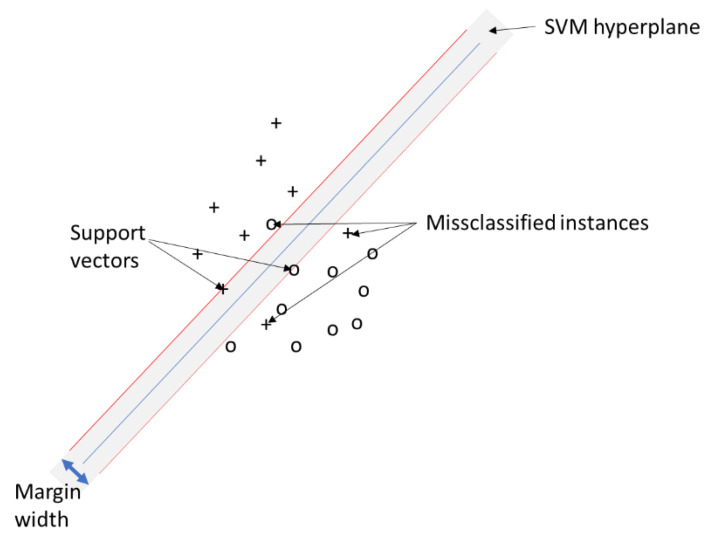
Linear SVM example adapted from [34].

**Figure 5 sensors-22-05683-f005:**
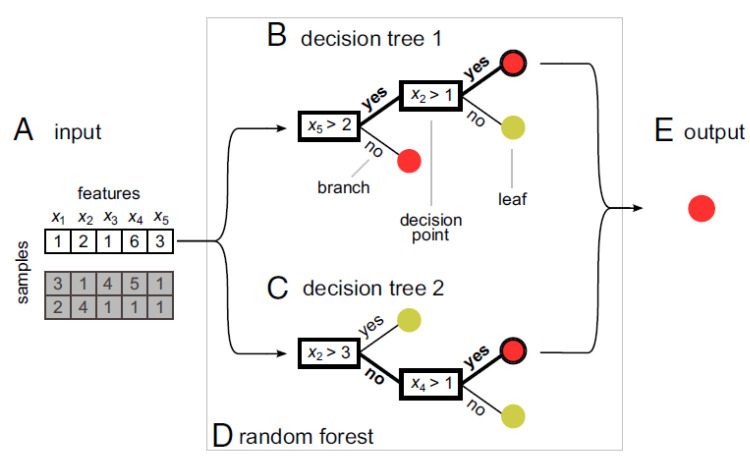
RF concept adapted from [37].

**Figure 6 sensors-22-05683-f006:**
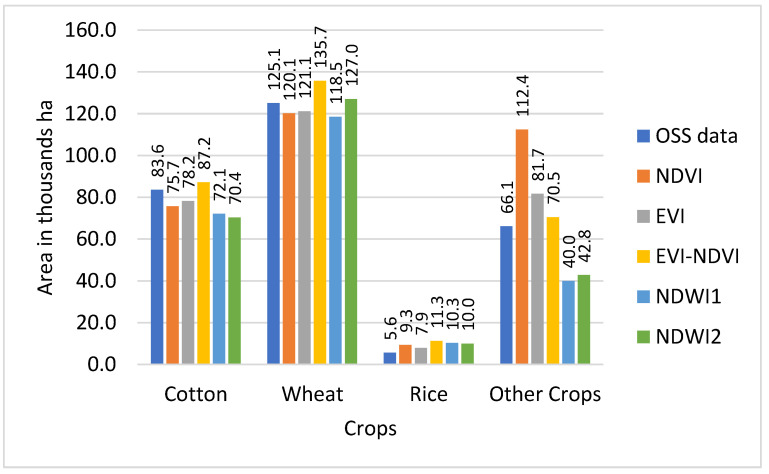
Comparison of classified crop types area with OSS data for RF classifier with S2 sensor.

**Figure 7 sensors-22-05683-f007:**
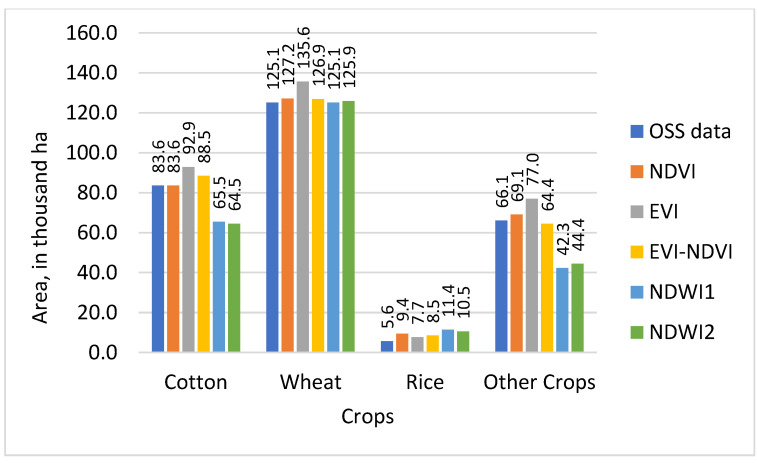
Comparison of classified crop types area with OSS data for SVM classifier with S2 sensor.

**Figure 8 sensors-22-05683-f008:**
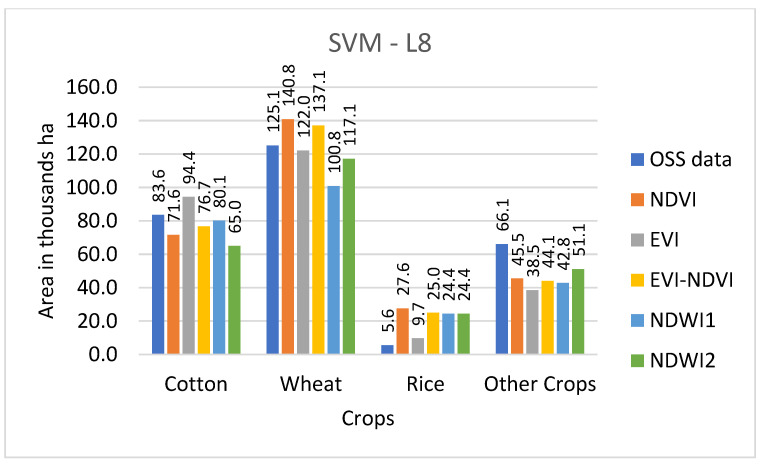
Comparison of classified crop types area with OSS data for SVM classifier with L8 sensor.

**Figure 9 sensors-22-05683-f009:**
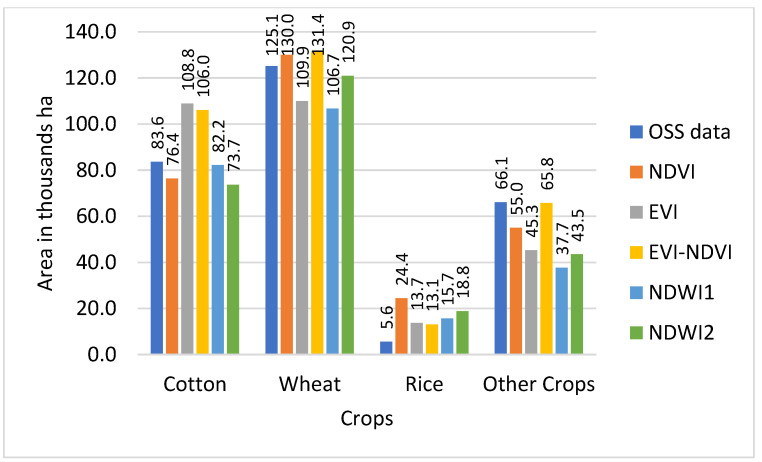
Comparison of classified crop types area with OSS data for RF classifier with L8 sensor.

**Figure 10 sensors-22-05683-f010:**
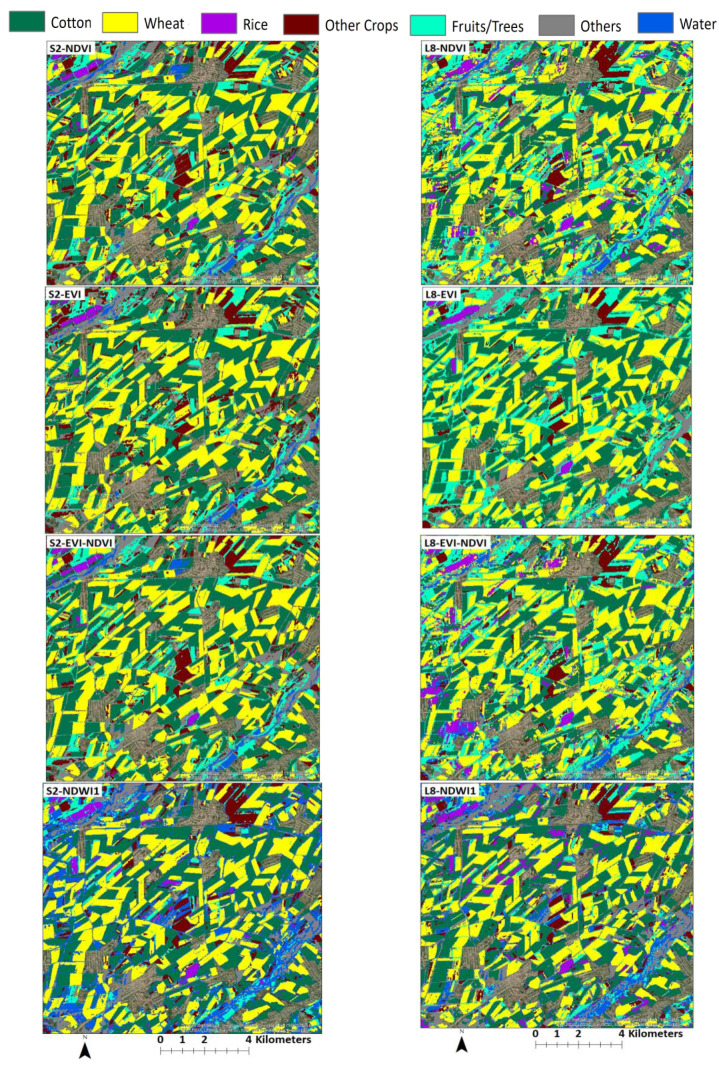
Subsets of land use maps derived from different indices using the SVM classification method for both S2 (**left**) and L8 (**right**).

**Figure 11 sensors-22-05683-f011:**
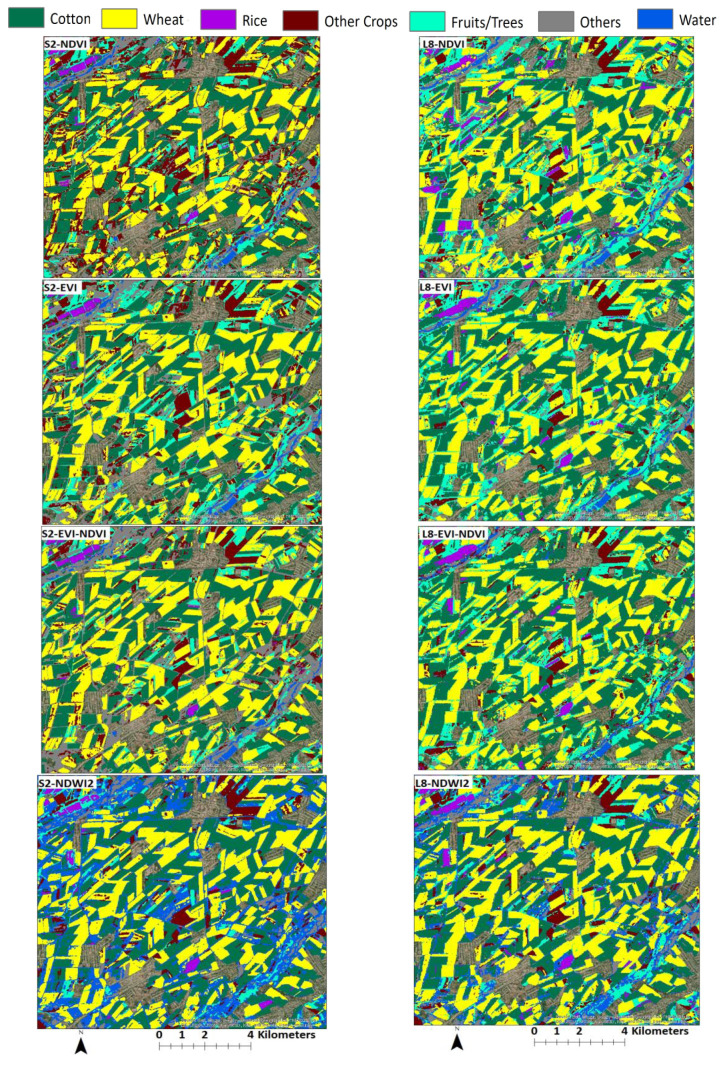
Subsets of LU maps derived from different indices using the RF classification method for both S2 (**left**) and L8 (**right**).

**Figure 12 sensors-22-05683-f012:**
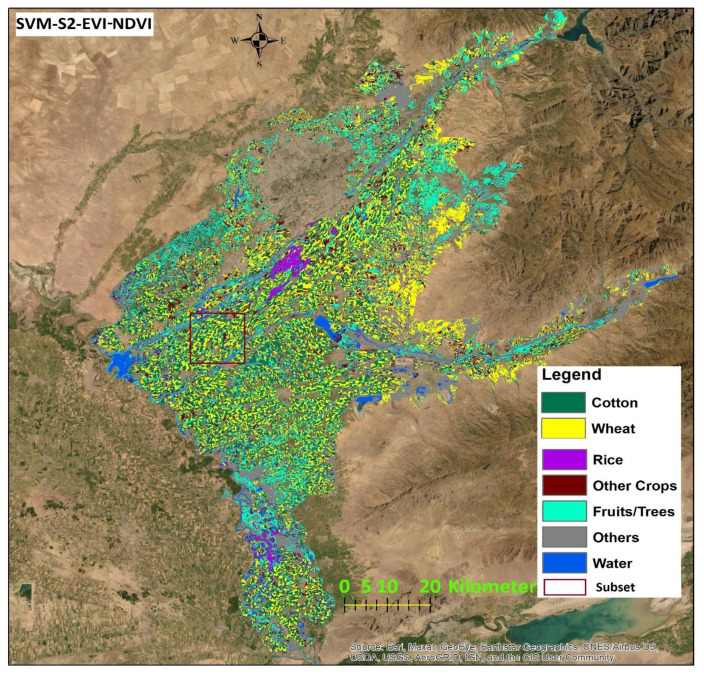
Classification result of S2-SVM-EVI-NDVI (highest OA 88% for SVM).

**Figure 13 sensors-22-05683-f013:**
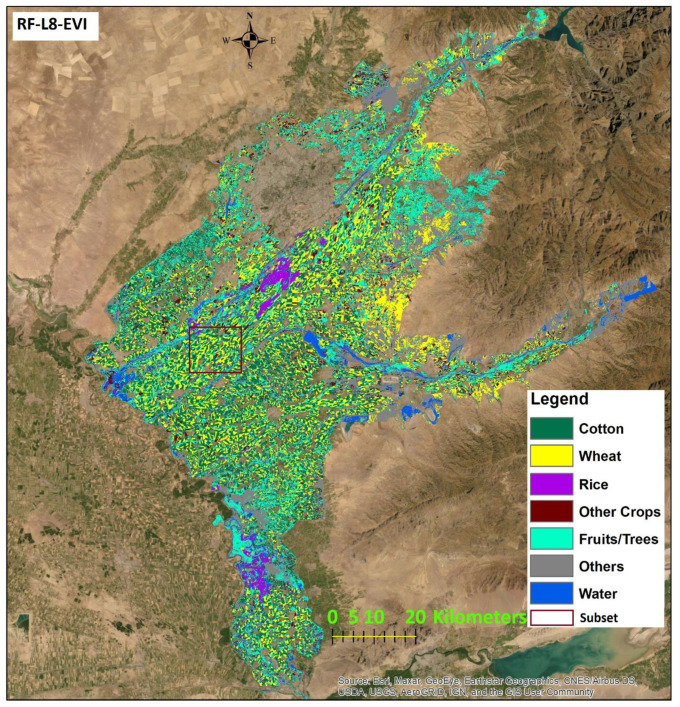
Classification result of L8-RF-EVI (highest OA 90% for RF).

**Figure 14 sensors-22-05683-f014:**
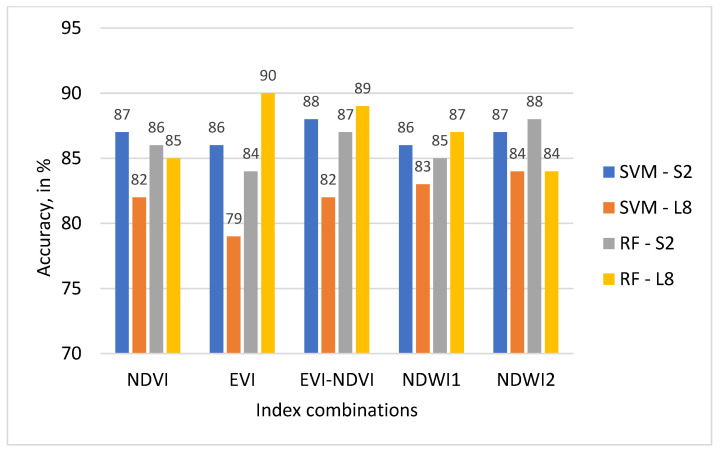
OA of SVM and RF classifiers on different indices and satellites.

**Figure 15 sensors-22-05683-f015:**
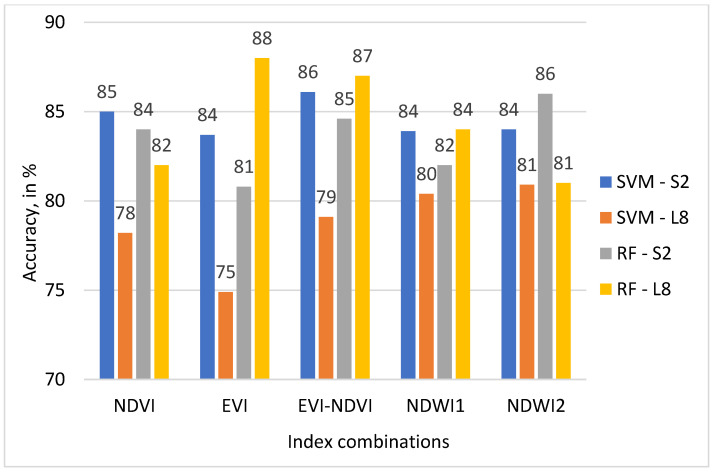
KA of SVM and RF classifiers on different indices and satellites.

**Figure 16 sensors-22-05683-f016:**
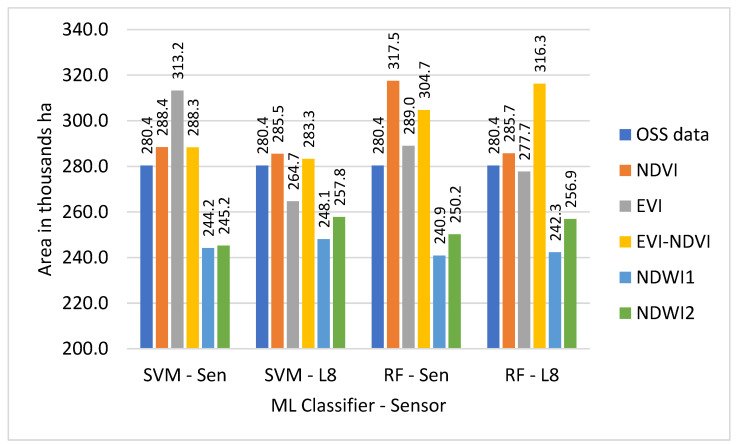
Difference between the total area of OSS data and classified area by sensors, classifiers, and indices.

**Table 1 sensors-22-05683-t001:** S2 and L8 tiles were downloaded for the classification.

Month	S2 Multispectral Imaging (MSI) Date				L8 Operation Land Imager (OLI) Date		
	T42TVL	T42TWK	T42TWL	T42TWM	153/031	154/031	154/032
May	25	7	7	7	28	3	3
June	24	6	6	6	13	20	20
July	9	1	1	1	15	22	22
August	3	5	5	5	16	7	7
September	2	4	4	4	1	8	8
October	2	4	4	4	3	26	26

**Table 2 sensors-22-05683-t002:** Specifications of spectral bands for S2 MSI [25] and L8 OLI [26].

Band Number	S2 MSI			L8 OLI		
	Description	Wave-Lengths (nm)	Spatial Resolution (m)	Description	Wave-Lengths (nm)	Spatial Resolution (m)
1	Coastal aerosol	433–453	60	Coastal aerosol	433–453	30
2	Blue	458–523	10	Blue	450–515	30
3	Green	543–578	10	Green	525–600	30
4	Red	650–680	10	Red	630–680	30
5	Vegetation Red Edge 1	698–713	20	NIR	845–885	30
6	Vegetation Red Edge 2	733–748	20	SWIR 1	1570–1650	30
7	Vegetation Red Edge 3	773–793	20	SWIR 2	2100–2300	30
8	Near-Infrared (NIR)	785–900	10	Panchromatic	500–680	15
8a	Narrow NIR	855–875	20			
9	Water vapor	935–955	60	Cirrus	1360–1390	30
10	SWIR-Cirrus	1360–1390	60	Thermal Infrared (TIRS) 1	10,600–11,200	100
11	SWIR 1	1565–1655	20	Thermal Infrared (TIRS) 2	11,500–12,500	100
12	SWIR 2	2100–2280	20			

**Table 3 sensors-22-05683-t003:** Numerical information on training and validation data.

Land Use	Training (Polygons/Pixels)	Validation (Pixels)
Cotton	150/2528	250
Wheat	300/5750	500
Rice	150/3038	250
Other Crops	150/2248	250
Fruits/Trees	150/3642	250
Bare land	150/2560	250
Others	150/2601	250
Water	150/2899	250

**Table 4 sensors-22-05683-t004:** Differences between classification results and OSS (1000 ha).

**S2-SVM**						
		NDVI	EVI	EVI-NDVI	NDWI1	NDWI2
Cotton		0	−9.3	−4.9	18.1	19.1
Wheat		−2.1	−10.5	−1.8	0	−0.8
Rice		−3.8	−2.1	−2.9	−5.8	−4.9
Other crops		−3	−10.9	1.7	23.8	21.7
	AM	−2.2	−8.2	−2.0	9.0	8.8
	WA (OSS):	−0.4	−2.5	−0.5	2.7	2.6
**S2-RF**						
		NDVI	EVI	EVI-NDVI	NDWI1	NDWI2
Cotton		7.9	5.4	−3.6	11.5	13.2
Wheat		5	4	−10.6	6.6	−1.9
Rice		−3.7	−2.3	−5.7	−4.7	−4.4
Other crops		−46.3	−15.6	−4.4	26.1	23.3
	AM	−9.3	−2.1	−6.1	9.9	7.6
	WA (OSS):	−1.6	−0.1	−1.7	31	2.1
**L8-SVM**						
		NDVI	EVI	EVI-NDVI	NDWI1	NDWI2
Cotton		12	12	−10.8	6.9	3.5
Wheat		−15.7	−15.7	3.1	−12	24.3
Rice		−22	−4.1	−19.4	−18.8	−18.8
Other crops		20.6	27.6	22	23.3	15
	AM	−1.3	4.9	−1.3	−0.2	6.0
	WA (OSS):	0.2	0.7	0.7	0.5	3.8
**L8-RF**						
		NDVI	EVI	EVI-NDVI	NDWI1	NDWI2
Cotton		7.2	7.2	−25.2	−22.4	1.4
Wheat		−4.9	−4.9	15.2	−6.3	19
Rice		−18.8	−8.1	−7.5	−10.1	−13.2
Other crops		−1.4	3.8	−4.3	−2.6	7.5
	AM	−1.35	3.75	−4.3	−2.6	7.45
	WA (OSS):	0.6	1.2	−0.2	−0.7	3.5

**Table 5 sensors-22-05683-t005:** Accuracy assessment (in %) results of SVM classifier.

Classes	NDVI				EVI				EVI-NDVI				NDWI1				NDWI2			
	UA		PA		UA		PA		UA		PA		UA		PA		UA		PA	
	S2	L8	S2	L8	S2	L8	S2	L8	S2	L8	S2	L8	S2	L8	S2	L8	S2	L8	S2	L8
Cotton	84	75	94	73	74	85	96	92	87	78	97	81	93	86	92	82	90	90	90	85
Wheat	87	85	92	81	86	78	92	89	88	82	91	83	92	93	93	90	95	89	91	89
Rice	93	75	78	68	97	94	70	94	95	83	80	74	90	83	92	85	92	84	88	87
Other crops	76	79	80	86	84	86	80	60	78	78	80	76	87	80	86	72	84	78	89	72
Fruits/trees	87	81	88	89	81	69	87	88	86	81	89	88	71	76	78	80	73	83	74	81
Others	89	84	83	85	92	65	75	86	89	91	83	82	92	87	82	90	94	85	86	83
Water	97	88	95	92	98	99	98	31	98	85	96	93	76	72	75	80	76	74	87	83
AM	88	81	87	82	87	82	85	77	89	83	88	82	86	82	85	83	86	83	86	83
**OA-S2**	87				86				88				86				87			
**OA-L8**	82				79				82				83				84			
**KA-S2**	85				84				86				84				84			
**KA-L8**	78				75				79				80				81			

**Table 6 sensors-22-05683-t006:** Accuracy assessment (in %) results of RF classifier.

Classes	NDVI				EVI				EVI-NDVI				NDWI1				NDWI2			
	UA		PA		UA		PA		UA		PA		UA		PA		UA		PA	
	S2	L8	S2	L8	S2	L8	S2	L8	S2	L8	S2	L8	S2	L8	S2	L8	S2	L8	S2	L8
Cotton	81	77	91	84	77	88	92	95	77	89	94	95	94	90	95	90	93	90	94	90
Wheat	91	90	87	84	86	94	89	92	91	92	93	87	92	94	92	92	94	89	93	90
Rice	94	83	76	76	96	94	68	97	92	94	76	97	90	90	94	88	94	90	93	90
Other crops	72	83	85	84	70	87	78	85	84	76	78	79	88	86	78	77	90	85	84	71
Fruits/trees	82	81	86	89	81	78	85	81	81	85	81	87	70	76	76	83	76	79	81	84
Others	89	86	84	82	85	92	73	91	85	96	86	82	95	95	80	86	93	92	82	75
Water	97	91	97	98	97	93	97	90	99	89	97	99	65	75	74	86	75	68	86	85
AM	87	84	87	85	85	89	83	77	87	89	86	89	85	87	84	86	88	85	88	84
**OA-S2**	86				84				87				85				88			
**OA-L8**	85				90				89				87				84			
**KA-S2**	84				81				85				82				86			
**KA-L8**	82				88				87				84				81			

**Table 7 sensors-22-05683-t007:** Deviation (in %) between the total mapped crop areas and OSS data (280.4 thousand ha).

Classifier- Sensor	Indices Combination				
	NDVI	EVI	EVI-NDVI	NDWI1	NDWI2
SVM-S2	3	12	3	−13	−13
SVM-L8	2	−6	1	−12	−8
RF-S2	13	3	9	−14	−11
RF-L8	2	−1	13	−14	−8

## Data Availability

Not applicable.

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
