# Peer review of "Irrigated Crop Types Mapping in Tashkent Province of Uzbekistan with Remote Sensing-Based Classification Methods"

_sensors, 2022, doi:10.3390/s22155683_

Round 1
Reviewer 1 Report
The manuscript aims at mapping several crop types based on remote sensing data from Sentinel 2 and Landsat 8 collected in 2018 (May-October). To this end, two machine learning algorithms are used, namely random forest and support vector machine. Uzbekistan’s Tashkent province is selected as the target area. Although the topic is interesting and the manuscript is generally well structured, several critical flaws (related specifically to content, coherence, and language) need to be addressed. I would recommend a “major revision” for now. A detailed review is attached.

Author Response
Response to Reviewer 1 Comments
GENERAL COMMENTS
Point 1: Coherence: ideas in many sections are incoherent;
Response 1: Coherence ideas are revised or removed based on your comments.
Point 2: The authors often use long sentences that make it difficult to understand their meanings. The reviewer recommends using short sentences for better understanding. Examples of such sentences are in lines #12, #57, #63, #77, #86, #102, …, #439
Response 2: The sentences in these lines revised.
Point 3: There are many language-related issues (not highlighted in this report unless necessary).
Response 3: The manuscript is revised concerning language-related issues.
Point 4: Double-check acronyms in the whole manuscript following these rules:
➢ Full names should be provided the first time acronyms are used. For instance, MODIS, AM, WA; ➢ Full names should be provided only once in the text. See, for instance, AI in line #130 which was already defined in line #77;
➢ Acronyms need to be provided only if they are used later on. For example, acronyms “LULCF”, “SFDI” and “SSR” were mentioned only once in lines #51, #115-116, and #161, respectively.
Response 4: Acronyms checked and corrected.
Point 5: A brief paragraph should be added to introduce the content of each section.
Response 5: A brief paragraph is added.
INTRODUCTION
Line #57: “This is due to …” I believe the authors here are referring to Uzbekistan and not necessarily the “large parts of the world”? Please elaborate.
The sentences are re-written.
Line #69: RSI? Do you mean RS?
RS imagery. It is corrected.
Line #72: “Similarly” here does not make much sense because a reference was not mentioned in the previous sentence.
It is a continuation of the previous sentence. The paragraph is moved.
Line #72: Reference [11] should be cited properly by specifying the name(s) of the authors. See also references [15] in Line #94. Authors are required to check all cited references in the manuscript.
The references are cited by names.
Line #91: The reviewer believes the authors need to be specific on what their review is about. The term “research interests and approaches” is too vague.
The sentence is re-written.
Line #130: Related to AI, the use of the term “machine learning (ML)” is more appropriate in the context of the present study.
ML is used instead of AI as recommended.
Line #146: It would be better to use the term “RS-based ML classifiers” instead of “RS classifiers” as SVM and RF are two ML classifiers. In this study, RS data were used as inputs.
The term “RS-based ML classifiers” is used instead of “RS classifiers” as recommended.
Lines #152-153: “Furthermore, given the fact that LS data have been available for the last five decades, results should also allow a conclusion regarding time-series studies on the development of irrigated croplands”. The authors exclusively used Landsat 8 data and not other data from other Landsat satellites. The reviewer believes it is too early to jump to such conclusions.
Your point is right while the sentence is not rightly placed. It is removed.
Point 6. Overall, the authors try to answer the central question: “Are we capable of identifying different crop types using ML algorithms based on at least medium-resolution RS data?”. However, in my opinion, the introduction as a whole, in terms of structure and content, does not provide a clear image of the existing literature, novelty, and specific research questions:
- Previous relevant research on mapping crops has not been cited, instead, the authors highlighted some studies that focused on a “variety of research interests and approaches” related to single crop types: maize, wheat, and rice. Moreover, some parts of the introduction are irrelevant such as in lines #131- 133: “By combining higher …”. Since UAV data was not used in this study, it is no point to mention it here. It would be rather suitable to mention these aspects as a future direction of the research. Authors are required to provide a concise and well-structured introduction with a relevant literature review that is related to the general objective which is mapping (and comparing) crops using ML algorithms based on RS data;
- The novelty of the research should be clearly stated. In other terms, compared to previous studies, what is the added value of this study?
- General and specific research questions or objectives need to be mentioned.
Response 6: I have tried to edit and revise the section “Introduction”. Your remarks are useful. And I will be happy to have time to improve it again, in case you are not satisfied.
MATERIALS AND METHODS
Point 7: Subsection “Study Area”: the authors need to add a brief description of the weather, the temporal ranges of seasons in the region, as well as the cultivated crop types vis-à-vis weather and seasonal changes.
Response 7: Brief description of your points is added.
Line #173: The authors should specify the reason(s) why they selected the months from May to October. Is that related to crops’ growth?
It is specified in the abstract shortly.
Point 8: Term “state statistics”: this term is used repeatedly in the whole manuscript. However, the nature and the structure of this data have not been described. An overview should be added to the subsection “Data”.
Response 8: The description of state statistical data is explained in the Data subsection. And Official State Statistics (OSS) data is issued in the whole manuscript.
Table #1: What are the download times listed in Table 1? Are they referring to the number of images used? Also, the authors should explain the terms “T42TVL, …” and “153/031” for readers who are not familiar with these notations.
It is described in the subsection “Data”.
Table #1: Abbreviations of sensors MSI and OLI were not defined previously. They should be added in 2.2.
Abbreviations are added in Table 1.
Line #187: The collection of ground-truth samples is not well detailed. I imagine that using Google Earth images only would be a challenging task to determine the nature of crops especially those with quasi-similar shapes, colors, features, etc. Did the authors collect these samples based on their knowledge, local people, or other means? Please elaborate.
Your points are described in subsection “Data”.
Point 9: Moreover, the authors did not address the temporal distribution of samples. In other terms, did the authors collect these samples during one month or all over the selected months? Also, in some countries, farmers tend to switch the cultivation of their agricultural lands. Is that applicable in the study area? If so, how did the authors consider this issue while collecting samples?
Response 9: As described in subsection “Data” training and validation samples were taken based on the expert’s knowledge of phenology and cropping calendar in the study area. This means temporal distribution is considered during sample taking. Besides, in our study area, cropping is controlled by the central administration. The farmers can not decide on cultivation changes. The only possibility is that they can have a second crop after harvesting winter wheat which is in July or August (no state-order crops are sown at this time). But it is not included in the official cultivated crop area. We also did not consider this issue in our research due to its heterogeneity. We are planning to study this topic as well in the future.
Line #215: No need to provide the full name of SVM.
It is removed.
Line #224: “remote sensing” is already abbreviated as RS
It is corrected.
Line #239: No need to provide the full name of RF.
It is corrected.
Lines #277-279: Abbreviations should be used.
It is corrected.
Figure 5: Since images have been collected from different rows/paths, I assume the authors merged multiple images. This step should be added to Figure 5 depicting the methodology as well as a brief description in the text.
The task merging and subset to the study area are added in Figure 5.
Figure 5: The extraction of reflectance values is not depicted in Figure 5.
It is depicted in Figure 5.
Point 10: Furthermore, indexes used in the methodology are not detailed (or at least referenced). How they are calculated. For instance how EVI+NDVI was calculated? Why they are selected in the first place? How do they differ from each other? The authors should answer these questions and cite relevant literature.
Response 10: Indices calculation is cited in subsection “Methodology”. Therefore, calculations of these indices are not explained here. EVI+NDVI is not a calculation. It is using EVI and NDVI together as input data. It is described in the manuscript now.
Point 11: Overall, the reviewer thinks Section 2 needs to be improved in terms of structure and content. Structures, Subsection 2.5 should be introduced first after 2.1-study area and 2.2-Data. I would suggest the following structure:
2.1 Study Area
2.2 Data
2.3 Methodology: A brief description of all steps regarding the following subsections
2.3.1 Data Preprocessing
2.3.2 Indexes
2.3.3 ML Algorithms SVM RF
2.3.4. Accuracy assessment
Response 11: The structure of Section 2 is improved according to the reviewer's recommendation.
RESULTS
Line #303: “these are presented in Table 4”
The sentence is moved to the proper place.
Figures 6-9: What label does “Statistics” refers to? A description should be added to subsection 3.1.
It is Official State Statistics (OSS) Data. It is described in the manuscript and replaced.
Figures 6-9: the index “EVI+NDVI” in the methodology is labeled differently in the figures (i.e., EVINDVI, and EVINDVI in Figure 12). The same remark applies to the labels used afterward.
It is remarked in all manuscripts as EVI-NDVI.
Figures 8-9: y-axis is not labeled
the y-axis is labeled now.
Lines #304-305: The authors should elaborate on the following sentence: “For each classification method, the arithmetic means as well as a weighted average using state statistics area as weights were calculated”. Particularly what do they mean by “state statistics”?
It is about comparing derived RS-based crop type classes area with OSS data. AS Arithmetic average is not the proper explanation, we calculated the weighted average using OSS data as weights. Now it is changed everywhere as I mentioned here.
Table 4: The authors should provide the full names for “AM” and “WA (StS)” in the text as well as their equations preferably in the methodology section (text and Figure 5)?
The full names are provided and “WA (StS)” is changed to “WA (OOS)” which means OSS data is used as weights in WA calculations.
Figures 10-11: The authors should add where the selected area is located in the study region. In the text, they need to provide why this part was selected?
The location of the subset is shown in Figures 12 and 13 where the whole study area is shown. And the reason for this area selection is shortly described in the manuscript.
Subsection 3.2: Text should be put first at the beginning subsection
It is corrected.
Line 355: CA-sensor-method-CA?
It is corrected. We meant all sensor-ML algorithm-Index combinations.
DISCUSSION
Figure 6, Figure 7, Figure 8, and Figure 9 âž” Figures 6-9
It is removed because not necessarily written there.
Point 11: This section does not discuss the obtained results, rather it provides an extension of the previous section “Results”. This is apart from some paragraphs that discussed the results notably the last two. Hence, the authors are invited to enrich the discussion section by comparing the results with previous studies, highlighting novelty, detailing theoretical and practical implications, pointing out limitations, and proposing future directions.
Response: The sections “Results” and “Discussion” are combined as “Results and Discussion”. We added some text and are ready to work further, in case you recommend revising it further.
Point 12: Looking at the abstract, the authors made a big claim that “since the launch of this satellite in 1972, historical irrigated cropland mapping is possible for the period up to today”. However, this is not discussed at all and the results that support such a claim can not be justified using one-year data from Landsat 8 which is more sophisticated than previous Landsat generations. The authors are required to revise such conclusions accordingly.
Response: The text you mentioned in your point is revised.
CONCLUSIONS
Point 13: The conclusions section is poorly written. Several key sentences do not make much sense. For instance:
- “As regards the difference between sensor data and state statistics area, the same conclusion needs to be drawn.”
- “It was found that the performance of both classifiers for rice crop is less reliable due to state statistics reports limitations and mixture of this class with second crops after winter wheat.”
Response 13: The conclusion section is re-written and your points were considered.
Point 14: The authors are invited to rewrite this section and professionally edit it.
Response 13: The section is revised and edited.
Reviewer 2 Report
The presented paper concern two remote sensing classification techniques including SVM and RF have been compared regarding their performance in deriving land use maps for different crop types using Landsat 8 and Sentinel-2 EO data. I found this paper very interesting where Several technical aspects were nicely implemented and explained sufficiently, and the authors showed a good level in both the theoretical and technical parts of the investigated subject. Certainly, the authors invested huge amount of time and have made a great effort to produce this high-quality of research which is clearly structured and the language used is largely appropriate. As final decision, I see that this manuscript in its form and level deserves to be accepted for publication in MDPI-SENSORS. Congratulations for all the authors.
Author Response
Response to Reviewer 2 Comments
GENERAL COMMENTS
Point 1: The presented paper concern two remote sensing classification techniques including SVM and RF have been compared regarding their performance in deriving land use maps for different crop types using Landsat 8 and Sentinel-2 EO data. I found this paper very interesting where Several technical aspects were nicely implemented and explained sufficiently, and the authors showed a good level in both the theoretical and technical parts of the investigated subject. Certainly, the authors invested huge amount of time and have made a great effort to produce this high-quality of research which is clearly structured and the language used is largely appropriate. As final decision, I see that this manuscript in its form and level deserves to be accepted for publication in MDPI-SENSORS. Congratulations for all the authors.
Response 1: Thank you very much for your attention and wishes. The manuscript is revised and the quality improved additionally.
Reviewer 3 Report
Dear authors,
Your paper, “Irrigated crop types mapping in Tashkent Province of Uzbekistan with remote sensing-based classification methods” requires some changes before it will be ready for publication.
My main problem is the main conclusion from this study: Landsat-8 and Sentinel-2 achieve the same results. Yet, it is because you used the same spectral indices for both sensors, and you did not utilize the advantage spectra as the thermal in Landsat and the red-edge in Sentinel-2. It will be of interest if you generate new two groups and run the classification and the accuracy process to see if any of these sensors will achieve better results because of their unique bands. These new groups could be including the red, NIR, SWIR1, TIR1 and TIR2 bands in Landsat-8 and the red, red-edge1-3, NIR and SWIR1 bands in Sentinel-2. It is not mandatory, but for your decision.
Title of the paper - Why do you think wheat is irrigated? according to your description, the growth period for this crop is winter-spring, and after that vegetable and melon are seeded (which make sense it is irrigated). If you do not have a reference to answer this question, please remove the word “Irrigated”, so the title will be “Crop type mapping…..”.
The phrase or term “ground-truth”, which you used to define the validation samples in Line 187, is misleading. How do you know what is the truth? A better phrase for this paper is validation samples or validation pixels. Further, please explain: (1) how did you know from historical maps where is a cotton or wheat field? and (2) which images (date, pixel size and sensor if possible) you used. This issue is also in Line 284, where you wrote “A subset from ground-truthing data was used to train SVM and RF classifiers”. Maybe it needs to be changed to “The training data (Table 3) was ….”?
The state statistics of the cropping area coverage – First, please explain in the Methods where you have these data, and what is this accuracy. Further, in the Discussion part, you mentioned several times there are problems with this dataset (Lines 406-408, 413-414) so please explain carefully why to use it and to what degree it can be trusted.
Your description of the Kappa, Lines 207-213 - please add a sentence to describe the importance of this matrix, as you did above for the other statistics. A good source is the Congalton paper, number 25 in your Reference list. It is important as it looks like you did not use this matrix as needed, for example in your interpretation of Figures 14 and 15 (Lines 46-422). My conclusions from these figures are that if Kappa is low, for example for SVM-L8, then a higher value of OA can not be trusted. On the other hand, RF-S2 and RF-L8 (both with EVI_NDVI) have Kappa>85 and OA>92, which show their accuracy.
Please explain how you convert Landsat-8 TOA to surface reflectance – you did write that it was performed in ArcMap and QGIS (Line 277) but not the method.
Figures 14-16 – First Figure 16 is presented before 14 and 15. Second, these figures should be presented in the Results part, and not in the Discussion
Editing and style –
- any fraction of a number, for example in Line 27 - 0,2 thousand ha, should be marked with a point. So, it will be “0.2 thousand ha”
- Lines 26 and 27 – numbers 14 and 15 should be removed
- Lines 48 and 120 – capital letter in the middle of the sentence
- Line 68 – what do you mean by “(i.e. no to low cloud cover)”? maybe change to (high cloud cover)
- Line 69 – please explain what is RSI
- Line 147 – please define the acronym LU
- Line 196, Figure 2 - the word “Fuits” in the Legend should be Fruits?
- Table 4, the L8-RF section (last part of this table) – the NDWI1 values are in a different color
- Table 5 – instead of the average (AM), I think that standard deviation or the coefficient of variance (standard deviation/mean) will serve better to show the variability between the classes.
- Lines 406-408 – the sentence is not clear: “Because the study area is located in the upstream of water resources and it has more access to the water resources than other areas of the country.”
- Figures 14 and 15 – the title of these figures should mention that these values are only for the wheat class
Author Response
Response to Reviewer 3 Comments
GENERAL COMMENTS
Dear authors,
Your paper, “Irrigated crop types mapping in Tashkent Province of Uzbekistan with remote sensing-based classification methods” requires some changes before it will be ready for publication.
Point 1: My main problem is the main conclusion from this study: Landsat-8 and Sentinel-2 achieve the same results. Yet, it is because you used the same spectral indices for both sensors, and you did not utilize the advantage spectra as the thermal in Landsat and the red-edge in Sentinel-2. It will be of interest if you generate new two groups and run the classification and the accuracy process to see if any of these sensors will achieve better results because of their unique bands. These new groups could be including the red, NIR, SWIR1, TIR1 and TIR2 bands in Landsat-8 and the red, red-edge1-3, NIR and SWIR1 bands in Sentinel-2. It is not mandatory, but for your decision.
Response 1: Thank you very much for your recommendation and we found it is interesting to see the results of the combination you suggested. Our main attention was to compare two different satellite sensors with different classification methods and indices. But we planning to study further comparative analysis with different aspects in the future.
Point 2: Title of the paper - Why do you think wheat is irrigated? according to your description, the growth period for this crop is winter-spring, and after that vegetable and melon are seeded (which make sense it is irrigated). If you do not have a reference to answer this question, please remove the word “Irrigated”, so the title will be “Crop type mapping…..”.
Response 2: Despite calling winter wheat, it is irrigated. Winter wheat is sown in October/November and it is irrigated as soon as it is sown. Besides, in early spring March, April and May also it is irrigated. In general, it uses irrigation water during the vegetation period. Therefore, it is called Irrigated crops.
Point 3: The phrase or term “ground-truth”, which you used to define the validation samples in Line 187, is misleading. How do you know what is the truth? A better phrase for this paper is validation samples or validation pixels. Further, please explain: (1) how did you know from historical maps where is a cotton or wheat field? and (2) which images (date, pixel size, and sensor if possible) you used. This issue is also in Line 284, where you wrote “A subset from ground-truthing data was used to train SVM and RF classifiers”. Maybe it needs to be changed to “The training data (Table 3) was ….”?
Answer 3: Ground-truthing phrase changed to validation pixels. The study area practices mostly cash crops controlled by government. Therefore, based on the local expert’s knowledge on crop phenology google historical maps were used. For this reason, only crop growth period scenes are chosen. Your points are corrected. Google historical images date are added and writing ground-truthing changed to to the training data in whole text.
Point 4. The state statistics of the cropping area coverage – First, please explain in the Methods where you have these data, and what is this accuracy. Further, in the Discussion part, you mentioned several times there are problems with this dataset (Lines 406-408, 413-414) so please explain carefully why to use it and to what degree it can be trusted.
Answer 4. State Statistics means official sown area data given by state statistics office. It is statistical book published every year. It is a total sown area at provincial level. It does not give any information at spatial and field extent of crops. Therefore, we compared official total sown area by crop types with classified total crop types area in order to find remote sensing techniques to create crop types map at spatial and field level. The text in the manuscript is revised about your point.
Point 5. Your description of the Kappa, Lines 207-213 - please add a sentence to describe the importance of this matrix, as you did above for the other statistics. A good source is the Congalton paper, number 25 in your Reference list. It is important as it looks like you did not use this matrix as needed, for example in your interpretation of Figures 14 and 15 (Lines 46-422). My conclusions from these figures are that if Kappa is low, for example for SVM-L8, then a higher value of OA cannot be trusted. On the other hand, RF-S2 and RF-L8 (both with EVI_NDVI) have Kappa>85 and OA>92, which show their accuracy.
Answer 5. We added a short description in chapter 2.3. considering your points. Table 5 and Table 6 is created based on the confusion matrices. The all matrices were 20 which is impossible to put in the manuscript. You point was right OA of SVM and RF classifier (figure 14) had an error. It was corrected and replaced. Now it can be seen that the values are correlated correctly.
Point 6. Please explain how you convert Landsat-8 TOA to surface reflectance – you did write that it was performed in ArcMap and QGIS (Line 277) but not the method.
Answer 6. It is explained and re-written.
Point 7. Figures 14-16 – First Figure 16 is presented before 14 and 15. Second, these figures should be presented in the Results part, and not in the Discussion.
Answer 7. Figure numbers renamed by order. Chapters 3 and 4 are combined as single chapter as Results and Discussion.
Editing and style –
- any fraction of a number, for example in Line 27 - 0,2 thousand ha, should be marked with a point. So, it will be “0.2 thousand ha”
It is corrected.
- Lines 26 and 27 – numbers 14 and 15 should be removed
It is removed.
- Lines 48 and 120 – capital letter in the middle of the sentence
It is corrected.
- Line 68 – what do you mean by “(i.e. no to low cloud cover)”? maybe change to (high cloud cover)
The sentence changed to “cloud free”.
- Line 69 – please explain what is RSI
It is explained.
- Line 147 – please define the acronym LU
Acronym is defined.
- Line 196, Figure 2 - the word “Fuits” in the Legend should be Fruits?
It is corrected.
- Table 4, the L8-RF section (last part of this table) – the NDWI1 values are in a different color.
Colors of Table 4 are now in similar.
- Table 5 – instead of the average (AM), I think that standard deviation or the coefficient of variance (standard deviation/mean) will serve better to show the variability between the classes.
It is shown in Table 4. Besides for accuracy assessment we have OA and KA to interpret the performance. Therefore, AM values are removed from Table 5.
- Lines 406-408 – the sentence is not clear: “Because the study area is located in the upstream of water resources and it has more access to the water resources than other areas of the country.”
The sentence is re-written.
- Figures 14 and 15 – the title of these figures should mention that these values are only for the wheat class
These figures represent the overall accuracy and kappa accuracy of all combinations which includes all classification classes.
Reviewer 4 Report
The manuscript contributes to the Sensors journal, however is important to add some comments that I already did to the manuscript body
In Figure 2 in the legend please write Fruits instead of Fuits
In suggest that Figure 5 send to the end of methodology section
Please write the same title in Y axis in both Figures 6 and 7
In Figures 8 and 9 add title in Y axis
In all manuscript body please homogenize the percent near to number (88%)
In the Discussion section please add two o more discussions with other researchers (compare findings)

Author Response
Response to Reviewer 4 Comments
GENERAL COMMENTS
The manuscript contributes to the Sensors journal, however is important to add some comments that I already did to the manuscript body
Point 1: In Figure 2 in the legend please write Fruits instead of Fuits
Response 1: It is corrected.
Point 2: In suggest that Figure 5 send to the end of methodology section
Response 2: Figure 5 is put to the end of methodology section.
Point 3: Please write the same title in Y axis in both Figures 6 and 7
Response 3: It is corrected.
Point 4: In Figures 8 and 9 add title in Y axis
Response 4: It is corrected.
Point 5: In all manuscript body please homogenize the percent near to number (88%)
Response 5: It is correct in all manuscript.
Point 6: In the Discussion section please add two o more discussions with other researchers (compare findings)
Response 6: Discussion chapter is updated with additional writing.
Round 2
Reviewer 1 Report
The reviewer thanks the authors for addressing most of their comments and suggestions.
In this review, I will not focus on language-related issues although the manuscript suffers from many notably the newly added text. I believe the journal editing service will handle them.
Another point is that I suggest the authors be concise when responding by providing the exact number (s) of the line(s) where the changes are introduced in the newly revised manuscript. That would make it easy for reviewers to follow and verify the modifications.
While the current draft is better, some minor questions remain:
In the R1 review, I did not suggest combining the “Results” and “Discussion” sections as the authors did mention in their response. In the previous manuscript, the authors included results in the “Discussion” section. Thus, I suggested moving these findings into the “Results” section and enriching the “Discussion” section.
As suggested in my previous review, the “Discussion” section should discuss the following points in separate paragraphs:
- A summary of your notable findings
- A comparison with previous studies highlighting whether your results are in agreement or not
- Theoretical and practical implications (e.g., for researchers, farmers, and decision-makers, …)
- Limitations: for example, compared to radar data
- Future directions: for example, in the conclusion, the authors mentioned that Landsat 9 could offer more opportunities (…), however, this was not mentioned in the discussion!
Author Response
Dear Reviewer,
Thank you very much for your comments. Your comments helped me a lot to improve the quality of manuscript. At the same time, I am learning a lot through revising the paper based on your comments.
I have re-written the whole Chapter "Discussion" according to your comments.
With best regards,
Elbek Erdanaev
